# TNF Superfamily and ILC2 Activation in Asthma

**DOI:** 10.3390/biom14030294

**Published:** 2024-02-29

**Authors:** Takahiro Matsuyama, Brittany Marie Salter, Nahal Emami Fard, Kentaro Machida, Roma Sehmi

**Affiliations:** 1Respiratory Research Group, Department of Medicine, McMaster University, Hamilton, ON L8S 4L8, Canada; matsuyat@mcmaster.ca (T.M.); brittmsalter@gmail.com (B.M.S.); emamifan@mcmaster.ca (N.E.F.); 2Department of Pulmonary Medicine, Graduate School of Medical and Dental Sciences, Kagoshima University, Kagoshima 890-8520, Japan; machida@m.kufm.kagoshima-u.ac.jp

**Keywords:** airway autoimmune response, eosinophilic asthma, GITRL, group 2 innate lymphoid cell, OX40L, RANKL, TL1A, TNF, TNFSF

## Abstract

Eosinophilic asthma is the most prevalent and well-defined phenotype of asthma. Despite a majority of patients responding to corticosteroid therapy and T2 biologics, there remains a subset that have recurrent asthma exacerbations, highlighting a need for additional therapies to fully ameliorate airway eosinophilia. Group 2 innate lymphoid cells (ILC2) are considered key players in the pathogenesis of eosinophilic asthma through the production of copious amounts of type 2 cytokines, namely IL-5 and IL-13. ILC2 numbers are increased in the airways of asthmatics and with the greatest numbers of activated ILC2 detected in sputa from severe prednisone-dependent asthma with uncontrolled eosinophilia. Although epithelial-derived cytokines are important mediators of ILC2 activation, emerging evidence suggests that additional pathways stimulate ILC2 function. The tumor necrosis factor super family (TNFSF) and its receptors (TNFRSF) promote ILC2 activity. In this review, we discuss evidence supporting a relationship between ILC2 and TNFSF/TNFRSF axis in eosinophilic asthma and the role of this relationship in severe asthma with airway autoimmune responses.

## 1. Introduction

Asthma is a complex airways disease, characterized by airway inflammation [1], affecting more than 330 million people worldwide [2]. The clinical presentation of asthma includes reversible airway obstruction, airway hyperresponsiveness, and inflammation of the airways [1]. This disease can be classified into various inflammatory phenotypes with the most well-defined being eosinophilic asthma [3]. Current treatment strategies aim to control airway inflammation mainly associated with eosinophilia and contribute to a decrease in symptoms and exacerbation rates for the majority of patients. However, more than 10% of asthma patients remain uncontrolled despite high-dose corticosteroid treatment with sustained airway eosinophilia [4]. Although recently licensed anti-type 2 biologics have significantly improved disease control, severe eosinophilic exacerbations persist in 30–50% of patients [5,6]. Identifying novel pathways driving airway eosinophilia could unveil new therapeutic targets for the treatment of difficult to control and therapy-resistant asthma. 

Group 2 innate lymphoid cells (ILC2) are lineage-negative cells that produce copious quantities of type 2 cytokines, in particular, IL-5, IL-9 and IL-13. ILC2 are primarily activated by epithelial-derived cytokines, such as IL-25, IL-33, and thymic stromal lymphopoietin (TSLP), providing an early and immediate innate cellular source of type 2 cytokines that initiate type 2 inflammatory response. Accumulating evidence suggests that additional mediators, aside from alarmins, which are proinflammatory endogenous molecules that mobilize and activate immune cells after cell injury, death, or immune induction response, promote ILC2 activity [7]. A growing body of evidence demonstrates that members of the tumor necrosis factor superfamily (TNFSF) provide important key co-stimulatory signals that control proliferation and effector function of numerous immune cells, including ILC2 [8,9]. In this review, we discuss the relationship between ILC2 and the TNFSF in the context of eosinophilic asthma. 

## 2. Role of ILC2 in Eosinophilic Asthma and Chronic Rhinosinusitis

Innate lymphoid cells are lympho-mononuclear cells that lack receptors for major immune cell lineages, including T and B cells [10]. These cells lack the recombinase activating gene, do not express antigen recognition receptors, and are not activated in an antigen specific manner. The ILC family is sub-grouped into distinct subsets (ILC1,2 and 3) based largely on expression of lineage specific transcription factors and cytokine profile [11]. ILC2 produces substantial amounts of IL-5 and IL-13 and, to a lesser extent, IL-4, IL-9, and amphiregulin when activated by alarmin cytokines (e.g., IL-25, IL-33, and TSLP), lipid mediators (e.g., leukotriene D_4_ and prostaglandin D_2_), neurotransmitters (acetylcholine), neuropeptides (neuromedin U, calcitonin-gene-related peptide, vasoactive intestinal peptide), aryl hydrocarbons, and cytokines (IL-2, IL-4, IL-7) [12]. ILC2 are mostly tissue resident cells found at barrier interfaces of the respiratory system, where they provide an early source of type 2 cytokines that initiate type 2 immune responses manifesting as eosinophilic inflammation and mucus hypersecretion in response to common aeroallergens, pollution, or viral or fungal agents [10,12]. In addition, expression of MHC-II by ILC2, with its capability for peptide-mediated activation of antigen-specific T cells, indicates that, by bridging innate and adaptive immune pathways, ILC2 play a fundamental role as drivers of tissue eosinophilia in atopic and non-atopic asthma [13,14,15,16]. More recently, mouse lung ILC2 have been shown to express the checkpoint inhibitor molecule, program cell death ligand-1 (PD-L1) up-regulated by IL-33, both in vivo and in vitro [17]. The study showed that PD-L1/PD-1 interactions between ILC2 and CD4^+^ T cell in the lung regulated T cell fate by promoting T helper 2 (Th2) lineage polarizations, as determined by the increased expression of GATA-3 and IL-13, leading to a more robust type 2 immune response. 

Increased numbers of ILC2 accumulate at sites of eosinophilic inflammation, including the upper and lower respiratory tracts. In stable allergic rhinitis patients, greater numbers of ILC2 are detected in peripheral blood and nasal mucosa compared to healthy controls [18,19], becoming rapidly elevated within 6 h following a local nasal allergen challenge [20,21], correlating with eosinophil and IL-5 levels [21]. Furthermore, blood ILC2 frequencies are increased during pollen season in those with seasonal allergic rhinitis and can be attenuated following antigen-specific immunotherapy [22]. In line with local residency of ILC2 in the upper airways, increased expression of TSLP and IL-25 within the nasal mucosa of patients with allergic rhinitis suggest that ILC2 activity may be driven by alarmin cytokines [23]. With respect to chronic rhinosinusitis (CRS), ILC2 are detected in nasal polyp tissue in patients with eosinophil predominance, further increased in those with co-existing asthma and correlating with type 2 inflammation and worsening symptom scores [19,24,25,26,27]. Furthermore, ILC2 derived from nasal polyp tissue also have the potential to produce more IL-5 and IL-13 compared to ILC2 from the nasal mucosa of healthy controls [28]. Alarmins are detected in nasal polyps, with TSLP specifically being elevated in CRS with nasal polyp compared to the nasal mucosa of healthy controls [29]. Collectively, these studies suggest that ILC2 play a role in eosinophilic inflammation within the upper respiratory tract, thereby contributing to the pathogenesis of CRS with nasal polyps with and without concomitant eosinophilic asthma. The exact immuno-pathological pathways that trigger ILC2 activity within the upper airways remains to be clarified. Although alarmins are expressed by nasal mucosal and nasal polyp tissue, there may be additional mediators, including members of TNFSF that control ILC2 function in this context, as discussed later. 

In asthma, ILC2 can be enumerated by flow cytometry in asthmatic airways [30,31,32,33]. Increased numbers of ILC2 were first identified in the airways of subjects with severe asthma compared to disease controls [33] or mild asymptomatic asthma [30]. In the latter study, the greatest number of activated ILC2, identified as IL-5/IL-13^+^ ILC2, was detected in sputum from severe asthma patients with uncontrolled airway eosinophilia (>3% sputum eosinophils) despite high-dose corticosteroid therapy. In contrast, the level of CD4^+^ T cell activation was either comparable or lower than in subjects with mild asthma. This suggests that ILC2, in contrast to CD4^+^ T cells in these patients, are corticosteroid insensitive and that uncontrolled activation of ILC2 may drive the persistence of airway eosinophilia.

In a follow up study, we expanded on these findings to show that an inhaled allergen challenge in mild asthmatics resulted in significant up-regulation of activated ILC2 numbers in sputum within 24 h, which then declined to baseline by 48 h post-allergen inhalation challenge [31]. This was associated with a significant decrease in blood ILC2 within the same time course. Interestingly, airway eosinophilia correlated with sputum IL-5^+^ ILC2 numbers post-allergen and sputum IL-33 levels. Therefore, allergen exposure results in activation of ILC2 within the local environment of the lower airways as opposed to systemic circulation. More recently, we showed that post-allergen challenge activated ILC2 is significantly increased within the airways of mild asthmatics within 7 h compared to a diluent challenge [34]. In agreement with our work, transcriptomic analyses of ILC2 in blood and bronchoalveolar lavage fluid (BALF) following segmental allergen challenges have shown that ILC2 are activated only within the airways of mild asthmatics. This was associated with the up-regulation of gene expression for IL-13 and the IL-33 receptor, IL-1RL1. In addition, CXCL12 and PGD_2_ in BALF negatively correlated with decreases in ILC2 numbers in the blood, indicating that these mediators may have the capacity to induce migration and lung-homing of ILC2 [35]. Therefore, ILC2 are rapidly and transiently activated within the airways over a discrete time course (significantly increased post-allergen at 7 h, maximal at 24 h, and reduced to baseline by 48 h). In severe asthma requiring high-dose inhaled corticosteroid therapy, ILC2 are the predominant type 2 cytokine-producing cells in the airways and may be an important contributor to symptoms in corticosteroid therapy resistant asthma. 

## 3. Role of ILC2 in Corticosteroid Resistance in Asthma

The relationship between ILC2 and corticosteroid insensitivity is an area of growing interest. There are conflicting reports with respect to whether ILC2 can respond to corticosteroids, although evidence suggests that local tissue micro-environmental cues and the degree of disease severity may influence the corticosteroid sensitivity of ILC2. 

In the upper airways, ILC2 are present within nasal polyp tissue of CRS patients with eosinophilia and are reduced by 50% following treatment with oral corticosteroids [26]. ILC2 are also detected in the nasal mucosa tissue of patients with allergic rhinitis and mild asthma when stable and increased 24 h post-nasal allergen challenge. Treatment with intranasal corticosteroid for one month compared to placebo significantly attenuated total IL-5/IL-13^+^ and HLA-DR^+^ ILC2 numbers 24 h post-nasal allergen challenge [36]. This was associated with an improvement in upper airway symptom scores and nasal function. In agreement with this, Yu et al. reported that inhaled corticosteroid treatment for three months significantly reduced blood ILC2 levels and that this was associated with a decrease in serum IL-13 levels and improvements in lung function and asthma control in patients with both moderate asthma and asthma with allergic rhinitis [37]. In addition, dexamethasone inhibited type 2 cytokine production by ILC2 in response to IL-25 + IL-33 stimulation by attenuating the mitogen-activated protein kinase (MEK)/Janus kinase (JAK)-signal transducer and activator of transcription (STAT) signaling pathways in vitro [37]. 

Studies by Liu et al. have shed further light on ILC2 resistance to corticosteroids [38]. Culturing blood ILC2 from poorly controlled moderate–severe allergic asthmatics stimulated with *Aspergillus* species or IL-2 + IL-33 in the presence of dexamethasone yielded a potent inhibition of IL-5 production, an effect that was reversed in the presence of IL-7 or TSLP in a MEK- and STAT-5-dependent manner. The inhibitory effects of dexamethasone were only observed with blood and not BALF-derived ILC2, suggesting compartmental differences of ILC2 sensitivity to corticosteroids in these patients. TSLP levels were increased in the BALF of these patients compared to disease control and demonstrated an inverse correlation with the inhibitory effect of dexamethasone on the respective BALF-derived ILC2 [38]. Additionally, dexamethasone treatment was shown to increase expression of IL-7Ra, a common receptor for IL-7 and TSLP in vitro. Increased IL-7Ra expression promoted heightened and sustained induction of pSTAT5 and MEK in ILC2 by TSLP with the proposal that this sustained induction of pSTAT5 and MEK promotes steroid resistance in ILC2 [38]. These findings are supported by transcriptomic studies that identified three genes responsible for corticosteroid resistance of ILC2, including CBX7, MEK2, and TLR2 [39]. It is postulated that pSTAT5 sequesters the glucocorticoid receptor in the cell cytosol, preventing nuclear translocation and alteration in pro-inflammatory cytokine generation. These findings were in agreement with an earlier study by Kabata et al., who reported that treatment with dexamethasone inhibited the production of IL-5 and IL-13 by CD4^+^ T cell but not ILC2 in an ovalbumin (OVA) + IL-33-induced murine model [40]. In this model, TSLP-neutralizing antibodies or STAT5 inhibitors restored corticosteroid sensitivity, indicating that TSLP drives corticosteroid resistance of ILC2 through a STAT5 dependent pathway [40]. 

In T cells, CD45RA is expressed on a naïve T cell, and CD45RO is expressed on activated and memory T cells [41]. ILC2 also expresses CD45RA and CD45RO, suggesting that resting ILC2 expresses CD45RA and that activated and inflammatory ILC2 express CD45RO. More recently, van der Ploeg et al. reported that patients with CRS with nasal polyps displayed altered ILC2 phenotypes with a predominance of ILC2 expressing CD45RO in both inflamed nasal polyp tissue and blood [42]. In the study using ILC2 from patients with CRS with nasal polyps, CD45RO^+^ ILC2 in nasal polyp exhibited greater levels of type 2 cytokine production in response to alarmin stimulation compared with CD45RA^+^ ILC2 in vitro. Furthermore, while CD45RA^+^ ILC2 was sensitive to inhibitory effects of dexamethasone, CD45RO^+^ ILC2 was largely unaffected [42]. CD45RO^+^ ILC2 is derived from resting CD45RA^+^ ILC2 upon activation by epithelial alarmins such as IL-33 and TSLP, which are tightly linked to STAT5 activation and up-regulation of the interferon regulatory factor 4 (IRF4)/B cell-activating transcription factor (BATF) transcription factors. CD45RO^+^ ILC2 are resistant to corticosteroids, the process of which is associated with metabolic reprogramming resulting in the activation of detoxification pathways, most notably glutathione metabolism. Promoting increased conjugation of glutathione to steroids, thereby facilitating increased elimination, provides a plausible mechanism for the intrinsic steroid resistance of CD45RO^+^ ILC2 [42].

With respect to the lower airways, as previously mentioned, we have shown that, in contrast to CD4^+^ T cell, total and activated numbers of ILC2 are present in higher numbers in the blood and sputum of prednisone-dependent severe asthmatics with uncontrolled eosinophilia compared to mild asthmatics. Furthermore, the frequencies of ILC2 positively correlated with eosinophils in blood and sputum, and fractionally exhaled nitric oxide, while the frequencies of ILC2 negatively correlated with predicted FEV1% [30]. Similarly, in the pediatric population, ILC2 is significantly increased in blood, BALF, and sputum of children with corticosteroid-resistant asthma versus disease control groups [43]. Assessing the level of responsiveness of ILC2 to corticosteroids in vitro, cultures of enriched blood ILC2 or CD4^+^ T cell from mild asthmatics stimulated with IL-2 and IL-33 produced significant levels of IL-5 and IL-13 that were attenuated in the presence of dexamethasone [31]. Thus, it seems that the degree of asthma severity dictates the level of corticosteroid responsiveness of ILC2. This theory is supported by Jia et al. and van der Ploeg et al., who showed that ILC2 phenotypes vary across degrees of asthma control and severity [42,44]. Blood ILC2 numbers are higher in asthmatics compared to healthy controls and produce greater levels of type 2 cytokines following stimulation with IL-25 and IL-33 in vitro. Interestingly, the degree of IL-13 expression in ILC2 was greatest in patients with uncontrolled asthma compared to subjects with partly or well-controlled asthma and healthy controls. Lastly, they showed that CD4^+^ T cells from healthy controls were more susceptible to the inhibitory effects of dexamethasone compared to ILC2 [44]. The frequencies of CD45RO^+^ ILC2 in blood positively correlated with the degrees of asthma symptom and exacerbation and the treatment dose of inhaled corticosteroids [42]. Therefore, CD45RO^+^ ILC2, through the resistance to steroids, may drive asthma severity. 

Collectively, these findings suggest that the inflammatory milieu within the airways can alter the ILC2 phenotype, in turn influencing the sensitivity of ILC2 to corticosteroid therapy. In all, ILC2 are responsive to corticosteroids in the upper airways of patients with mild allergic rhinitis, CRS with nasal polyps, and in mild asthmatics. However, as the severity of asthma progresses with uncontrolled eosinophilia, there is an altered tissue cytokine milieu that can alter the ILC2 phenotype, resulting in steroid insensitivity. With increased predominance of alarmins, in particular, TSLP, the tissue cytokine milieu can dictate an altered ILC2 phenotype and increased resistance to inhibitory effects of corticosteroids. The role that additional cytokines other than alarmins play in this process is discussed below.

## 4. Relationship between TNF Superfamily and ILC2

TNF was first discovered over 40 years ago as a mediator of fever and cachexia. It has been shown to have many important physiological and pathological actions. TNF causes tumor cell necrosis and apoptosis as well as host defense against bacterial and viral infection. Furthermore, TNF plays a central role in autoimmune diseases such as rheumatoid arthritis (RA), inflammatory bowel diseases (IBD), and systemic lupus erythematosus, where anti-TNF therapy has been clinically applied [45]. Currently, 19 additional structurally-related cytokines and 29 receptors have been identified, leading to the creation of TNFSF and the TNF receptor superfamily (TNFRSF) [46]. Members of the TNFSF provide key communication signals between immune cells, including T cells and NK cells [47,48]. Given the important role of ILC2 in asthma, whether TNFSF members can affect ILC2 function has been a question of interest. Although alarmins are thought to be key mediators of ILC2 activation, mouse models have shown that, in the absence of these alarmin cytokines or the cognate receptors, functional ILC2 persist, indicating the existence of additional stimulatory pathways that drive ILC2-mediated type 2 immune responses [49,50].

The current body of evidence reports that ILC2 express four distinct receptors for TNFSF, including TNFR2 (TNFRSF1B), a receptor activator of nuclear factor-κB (RANK; TNFRSF11A), glucocorticoid-induced TNF receptor (GITR; TNFRSF18), and death receptor 3 (DR3; TNFRSF25), which bind to their ligands/mediators TNF(TNFSF2), RANKL (TNFSF11), GITRL (TNFSF18), and tumour necrosis factor-like cytokine 1A (TL1A;TNFSF15), respectively [50,51,52,53]. These TNFSF–TNFRSF axes induce ILC2-derived type 2 cytokine production and ILC2 survival, which can be enhanced in the presence of epithelial-derived cytokines. Furthermore, OX40L (TNFRSF4) expressed on ILC2 provides T cell co-stimulation via OX40, which is expressed on T cells [54] (Figure 1). Studies showing that members of the TNFSF modulate ILC2 function are reviewed below (summarized in Table 1).

### 4.1. TNF/TNFR2 Axis and ILC2s

The most well characterized members of the TNFSF–TNFRSF system are TNF and its receptors, TNF receptor 1 and 2 (TNFR1; TNFRSF1A and TNFR2). TNF is a pro-inflammatory cytokine that plays a role in several inflammatory diseases, including asthma [64]. It is increased in the airways of severe asthma patients [65,66], and murine studies have confirmed the function of TNF in promoting bronchoconstriction and airway hyperresponsiveness (AHR) [67,68]. Although TNF-neutralizing antibodies are commonly used to treat patients with autoimmune diseases (i.e., RA, IBD, and psoriasis), the efficacy of these drugs with respect to human asthma is not shown [69]. Anti-TNF therapy is associated with an increased risk of infection and tumor progression due to the existence of two structurally different TNF receptors that have opposing functions. TNFR1 is expressed on most cell types and is associated with cell survival and proliferation as well as cytotoxicity [70]. In contrast, TNFR2 expression is restricted to selected cell types and is associated with cell survival and proliferation. This is because TNFR2 lacks the death domain signaling molecules that are present with TNFR1 [71]. The TNF–TNFR2 axis has been identified as a central checkpoint in modulating immune regulation. More specifically, regulatory T cells (Treg) express high levels of TNFR2 compared to conventional T cells and are crucial for Foxp3 stabilization, proliferation, and activation of Treg through NFκB-dependent intracellular signaling [72,73,74].

Until recently, it was largely unknown whether TNF activated ILC2 and whether ILC2 expressed TNFR1 or TNFR2. Ogasawara et al. [51] reported that ILC2 express TNFR2 but not TNFR1, at both the mRNA and the protein levels and that there is similar expression in the blood from healthy controls and from the nasal polyp tissue from patients with CRS with nasal polyps. Enriched blood and nasal polyp ILC2 cultured with TNF demonstrated a time- and dose-dependent production of IL-5 and IL-13, which was further enhanced in the presence of TSLP or IL-33 in vitro. Treatment with dexamethasone and an inhibitor of NFκB resulted in attenuation of the TNF-mediated IL-5 and IL-13 production. Thus, TNF-mediated effects on ILC2 are dependent on NFκB intracellular signaling [51] and greatly enhance type 2 cytokine production by ILC2. Hurrell et al. further confirmed these findings by showing that ILC2 selectively expressed TNFR2 at baseline and following stimulation with IL-33, whereas ILC2 did not express TNFR1 [55]. Stimulation with IL-33 up-regulated TNFR2 expression on ILC2 by 24 h, peaking at 72 h. Moreover, a blockade of the TNF–TNFR2 axis inhibited ILC2 survival and cytokine production as well as ILC2-dependent AHR. The effect of the TNF–TNFR2 axis on ILC2 survival and cytokine secretion was dependent on NFκB signaling. The TNF–TNFR2 axis on ILC2 activation observed in mice translated to human ILC2, where treatment with an NFκB inhibitor attenuated lung inflammation and AHR in mice receiving adoptively transferred human ILC2 activated with TNF-α [55]. Collectively, these findings demonstrate that specifically targeting the TNF–TNFR2 axis (as opposed to the TNF–TNFR1 axis) may be a novel therapeutic avenue for asthma and could reduce adverse effects associated with pre-existing anti-TNF agents. More studies, however, are needed to ascertain the exact relationship between the TNF–TNFR2 axis and ILC2 in human asthma. 

### 4.2. TL1A–DR3 Axis and ILC2

There is emerging evidence in both animal and human studies that another member of the TNFSF, known as TL1A, and its receptor, DR3, can modulate ILC2 function. Normally undetectable, TL1A is produced by stressed epithelial cells, activated dendritic cells, and macrophages at mucosal barrier interfaces and is a co-stimulatory molecule for T cells [75,76]. Genome-wide association studies (GWAS) report a significant association between single-nucleotide polymorphisms in TL1A and chronic IBD [77]. Moreover, in a rat model of IBD, a blockade of TL1A significantly reduced inflammatory cell infiltrate and fibrosis to ameliorate the increase in colon wall thickness and ulceration [78]. With regard to asthma, a blockade of TL1A with neutralizing antibodies suppressed airway inflammation and levels of IL-4, IL-5, and IL-13 in a murine model of asthma [78,79]. DR3-knock out (KO) mice also exhibited reduced cellular infiltration and goblet cell hyperplasia within the airways of these mice [80]. 

ILC2, in both mice and humans, have been shown to express DR3 [49,50,57]. Murine studies have shown that ILC2 numbers and type 2 cytokine production were increased following stimulation with TL1A in a DR3-dependent manner (responses not seen in DR3-KO mice) and a T/B cell-independent manner (normal responses to TL1A seen in Rag2-KO mice) in vivo [50]. Therefore, these findings suggest that the TL1A–DR3 axis does not influence the development of these cells but rather affects pro-inflammatory activity. Despite DR3-KO mice displaying normal responsiveness to administration of IL-25 or IL-33, these mice demonstrated impaired innate immune responses when infected with *N. brasiliensis* or when challenged intranasally with papain in vivo [50,56]. This impaired innate immune response may be attributed to the evidently lower ILC2 numbers and reduced IL-5 and IL-13 levels in BALF. Interestingly, DR3-KO mice showed no difference in type 2 cytokine expression by CD4^+^ T cell. These findings suggest that the TL1A–DR3 axis may be an important co-stimulatory pathway that predominantly activates ILC2 and optimizes type 2 immune responses. Furthermore, IL-13 production from TL1A-stimulated ILC2 promoted mucus secretion and lung fibrosis [58]. Thus, TL1A may contribute to the pathogenesis of not only asthma but also chronic obstructive pulmonary disease and cystic fibrosis. In addition, in metabolic disorders, DR3 stimulation was shown to improve glucose tolerance, protect the onset of insulin resistance, and attenuate established insulin resistance in an ILC2 function-dependent manner in a murine model of a high fat diet. Therefore, the DR3 agonist treatment may be also expected to be a novel therapeutic option for prevention and treatment of type 2 diabetes mellitus [57].

With respect to human studies, Yu et al. identified DR3 mRNA highly expressed in human ILC2, which was confirmed through flow cytometry using blood ILC2 from healthy controls [50]. They further showed that human ILC2 expanded with IL-2, IL-7, and IL-25 for 6 h resulted in up-regulation of DR3 at the mRNA level in vitro. In addition, TL1A alone in the absence of IL-2 or IL-7 was able to induce ILC2 to produce IL-5 and IL-13 [50]. Shafiei-Jahani et al. also demonstrated that human ILC2 stimulated with TL1A in the presence of IL-2, or IL-7 for 48 h stimulated IL-5 and IL-13 as well as granulocyte macrophage colony-stimulating factor (GM-CSF) secretion [57]. There are, however, few studies investigating the relationship between the TL1A–DR3 axis and ILC2 in human asthma. A recent report determined the level of TL1A and DR3 expression in the airways of asthmatics and how the TL1A–DR3 axis affected ILC2 activity in response to the inhaled allergen challenge [34].. There was an increase in DR3^+^ ILC2 numbers 24 h post-allergen in mild asthmatics. Similarly, there was an up-regulation in the total CD4^+^ T cell and DR3^+^ CD4^+^ T cells in sputum 24 h post-allergen. This was associated with increased airway TL1A levels at 24 h post-allergen but not the diluent challenge [34]. These findings suggest that ILC2 and CD4^+^ T cell responsiveness to the TL1A–DR3 axis becomes up-regulated within the airways following exposure to allergen. The interaction between these cells and the TL1A–DR3 axis may contribute to downstream type 2 inflammation. To further support these findings, we determined what signals can modulate the expression of DR3 on ILC2 and CD4^+^ T cells in vitro. Overnight stimulation with IL-2, IL-33, and TSLP, but not IL-25, yielded a significant increase in the proportion of blood ILC2 expressing DR3, however, this was not seen with CD4^+^ T cells [34]. These findings demonstrate that alarmins, specifically IL-33 and TSLP, have the capacity to up-regulate DR3 expression on ILC2, thereby enhancing TL1A-mediated stimulation of these cells. Expanding on this, we showed that stimulation of enriched blood with ILC2 with IL-2 alone or TL1A and IL-2 resulted in enhanced IL-5 expression by ILC2, which was reversed in the presence of either a TL1A-neutralizing antibody or dexamethasone in vitro. A combination of TL1A and TSLP (but not TSLP or TL1A alone) was able to induce IL-5 production by ILC2, however, this was resistant to the inhibitory effects of dexamethasone [34]. To our knowledge, this is the first study in humans that assessed changes in the TL1A–DR3 axis following allergen challenges in mild asthmatics and how this affected ILC2 function. Given the ability of TL1A–DR3 axis to mediate ILC2 activity, it may pose as a novel target for treatment of asthma. Notably, the combination of alarmins and TL1A conferred resistance to treatment with dexamethasone, suggesting that the TL1A–DR3 axis may have an additional role in corticosteroid resistance as a co-stimulatory pathway. 

### 4.3. RANKL–RANK Axis and ILC2

RANKL, as a member of TNFSF, and its receptor, RANK, play important roles in osteogenesis, organization of lymphoid tissues, as well as dendritic cell survival [81]. Dysregulation of the RANKL–RANK axis is involved in numerous inflammatory diseases, including RA. In clinical practice, an anti-RANKL monoclonal antibody is used for the treatment of osteoporosis and bone metastasis in addition to RA [82]. RANKL is an osteoclast differentiation factor expressed by osteoblasts that mediates the production of prostaglandin, TNF, IL-1, IL-17, and IL-6. In contrast, osteoprotegrin (OPG; TNFRSF11B) is expressed on osteoblasts and acts to competitively bind to RANKL as a decoy receptor, inhibiting RANKL binding to RANK. Thus, RANKL binding to OPG induces bone resorption by inhibiting osteoclast activation and differentiation [83,84,85]. More specifically, RANKL expression on T cells acts to induce survival and maturation of dendritic cells. A murine study has shown that OVA-induced airway inflammatory responses are reversed in the presence of OPG, suggesting that the blockade of the RANKL–RANK axis plays a role in asthma inflammation [86]. Another study also has shown that blockade of the RANKL–RANK axis results in attenuation of allergic airway inflammation in mice [87]. 

Bone marrow (BM)-derived ILC2 expressed with high levels of RANKL and blood ILC2 co-cultured with BM-derived monocyte/macrophage lineage cell (BMM) in the presence of IL-7 induced the differentiation of tartrate-resistant acid phosphatase-positive osteoclasts [59]. On the other hand, IL-33-stimulated BM ILC2 down-regulated RANKL expression and converted BMM differentiation into M2 macrophage-like cells rather than osteoclasts by the production of GM-CSF and IL-13 [59]. Thus, ILC2 play an important role in osteoclast activation and contribute to bone homeostasis. Furthermore, Ogasawara et al. assessed TNFSF expression in nasal polyp tissue from patients with CRS with nasal polyps [52]. RANK was expressed on ILC2 in human blood and nasal polyps. Stimulation with a RANK agonist induced production of type 2 cytokines from human ILC2, which was further enhanced in the presence of TSLP in vitro. They further showed that membrane-bound RANKL was detected on CD4^+^ T cells, and co-culture of these cells with ILC2 derived from nasal polyps resulted in enhanced type 2 cytokine production, which was attenuated in the presence of an anti-RANKL monoclonal antibody [52]. These findings demonstrate that the RANKL–RANK axis can induce ILC2 activity, further enhanced in the presence of TSLP. Unfortunately, no studies have been done thus far that have looked at the role of the RANKL–RANK axis in promoting ILC2 function in the context of human asthma. 

### 4.4. GITRL–GITR Axis and ILC2

GITRL and its respective receptor, GITR, represent another TNFSF axis that may have a potential role in asthma. GITR is expressed at steady-state and following stimulation on Tregs. In addition, GITR has further been detected in B cells, NK cells, granulocytes, and macrophages. The GITRL–GITR axis has the capacity to prevent the suppressive function of Treg and induce T cell proliferation as well as cytokine production [88]. Furthermore, GITR signaling can potentiate AHR through inducing type 2 inflammation in an OVA-induced murine model [89]. These responses are mediated via mitogen-activated protein kinases (MAPKs)/NFκB signaling in a house dust mite-challenged murine model of asthma [90]. A study by Zhang et al. reported the presence of GITR-positive cells in the spleen and lung tissue that was further increased in a murine model of asthma following an allergen challenge [53]. A significant positive correlation between GITR expression, ILC2-associated molecules (ST2, ROR-α), and type 2 cytokine expression (IL-5, IL-13) was found. GITR was further identified in ILC2 within lung tissue, and stimulation with GITRL resulted in increased numbers of lung ILC2 in vitro [53]. Galle-Treger et al. expanded on these findings to show that human and naive or IL-33-stimulated mouse ILC2 express GITR in vivo [60]. Similarly, ILC2 stimulated with IL-33 and a GITR agonist resulted in the increased production of type 2 cytokines in vitro. Furthermore, GITR induced the production of ILC2 effector cytokines as well as inhibiting ILC2 apoptosis by activating the NFκB pathway. Of note, GITR engagement improved ILC2-mediated glucose tolerance, thereby limiting the early onset of insulin resistance and as well as ameliorating established insulin resistance in a murine model of a high fat diet [60]. An additional study has shown that ILC2 numbers are lower in the lungs of GITR-KO mice compared to wild-type, and that ILC2 expresses lower levels of IL-5 and IL-13 mRNA. Furthermore, IL-5-dependent eosinophilia in BALF was reduced in GITR-KO mice [61]. It is therefore possible that the GITRL–GITR axis can activate ILC2 to promote airway eosinophilia or improve type 2 diabetes mellitus. So far, similar studies have not investigated the GITRL–GITR axis and ILC2 in subjects with asthma or type 2 diabetes mellitus. 

### 4.5. OX40L–OX40 Axis and ILC2

The TNFRSF member OX40L (TNFSF4) and the receptor OX40 are well known for being involved in co-stimulation of T cells [46]. Interestingly, ILC2 within the lungs has been shown to express high levels of OX40L upon stimulation with IL-33 [54]. In recent studies, ILC2 has been shown to activate T cells via OX40L–OX40 axis. Okuyama et al. demonstrated that ILC2 and CD8^+^ T cell in tumors of IL-33-treated mice express OX40L and OX40, respectively, and blockade of the OX40L–OX40 pathway suppressed the anti-tumor effects of IL-33 in the tumor microenvironment in vivo [62]. Co-culture of CD8^+^ T cells with IL-33-stimulated ILC2-induced cell activation and proliferation of CD8^+^ T cells. This effect was significantly suppressed by the administration of an anti-OX40L blocking antibody [62]. Thus, the IL-33–ILC2 axis promotes CD8^+^ T cell responses via the OX40L–OX40 axis to induce anti-tumorigenic effects. Furthermore, in the lungs of mice, infection with respiratory syncytial virus (RSV) induced an expansion and activation of CD4^+^ T cells, which depended on lung ILC2. RSV infection increased numbers of OX40^+^ CD4^+^ T cells and OX40L^+^ ILC2 in mouse lungs. Co-cultures of CD4^+^ T cells with ILC2 in the presence of an anti-OX40L antibody significantly reduced T cell cytokine production of IFN-γ, IL-2, and IL-13 [63]. These findings suggest that lung ILC2 may regulate RSV-induced CD4^+^ T cell expansion and activation via OX40L/OX40 interaction [63]. With respect to type 2 airway inflammation, co-culture of anti-CD3/CD28-stimulated CD4^+^ T cells with ILC2 resulted in significant increases in IL-4, IL-5, and IL-13 levels in the supernatants inhibited with a presence of an antibody to OX40L in vitro. On the other hand, in cultures of CD4^+^ T cells alone, the anti-OX40L antibody had no effect [16]. Additionally, alarmin-induced expression of OX40L on ILC2 provided tissue-restricted T cell co-stimulation that was essential for Th2 cell and Treg responses in the lung. In contrast, deletion of OX40L on ILC2 could attenuate the expansion of Th2 cell and Treg and allergen-induced type 2 airway inflammation [54]. These findings suggest that OX40L expression on ILC2 is up-regulated in response to alarmin stimulation and is a critical checkpoint for orchestrating adaptive type 2 inflammation. More studies are required to understand the importance of the OX40L–OX40 axis in mediating an interaction between T cells and ILC2 in asthma.

## 5. Role of TNF Superfamily and ILC2 in Airway Autoimmune Responses in Eosinophilic Asthma

Recent studies have described a subset of severe asthmatics that have airway autoimmune responses, manifesting as auto-IgG antibodies directed against cell-derived granule proteins, including eosinophil-granule derived eosinophil peroxidase (EPX), macrophage scavenger receptor with collagenous structure (MARCO), and nuclear/extranuclear antigens [91,92]. In total, 55% of patients with moderate to severe asthma had two or more autoantibodies, increased eosinophils in the airways, recurrent infections, and exacerbations of asthma [92]. It is proposed that, over time, chronic eosinophilic inflammation results in the accumulation of increasing levels of self-antigens. In agreement with this and prominent in the sputum of severe asthma patients were detectable IgG antibodies against EPX and other cellular components, including double stranded DNA and histones. In contrast, there were relatively lower titers of these antibodies in the peripheral circulation. Autoantibodies within the airways were associated with clinical markers of eosinophilic degranulation, including sputum EPX and free eosinophilic granules [91]. This suggests that accumulation of self-antigens creates an environment in the airways that may break down immune tolerance. It is possible that local signaling pathways in the airway may be promoting autoantibody generation [93]. For example, airway autoantibodies are associated with the formation of eosinophil extracellular traps, which were resistant to treatment with dexamethasone [91]. In addition, increased levels of anti-EPX IgG were observed in severe asthmatics who did not respond to anti-IL-5 therapy, and severe asthmatics with two or more autoantibodies, such as anti-EPX, MARCO, and nuclear/extractable nuclear antigens, were at a higher risk of exacerbations and were likely not to respond to biologics, as their titers were not attenuated by current anti-inflammatory treatment, regardless of eosinophil suppression [92,94,95]. Specifically, mixed granulocytic (>2% sputum eosinophils and >64% sputum neutrophils) asthma patients with anti-MARCO IgG antibodies had higher risk of subsequent asthma exacerbation triggered by infection. This may be because monocyte-derived macrophage treated with anti-MARCO IgG had reduced ability to phagocytose and clear bacteria [96]. Collectively, these findings suggest that the presence of airway autoimmune responses may not only confer corticosteroid resistance but can predict a higher failure rate to biologic therapy and a higher risk of asthma exacerbations. Understanding how autoimmune responses generate corticosteroid insensitivity/resistance is integral to developing future therapies for these patients. 

Our own recent findings suggest a link between airway autoimmune responses and the TNFSF in driving ILC2 activity and subsequent corticosteroid resistance [34]. Interestingly, the TNFSF has been found to be prevalent in autoimmune diseases including RA, IBD, and systemic lupus erythematous [97,98,99,100,101,102,103,104,105,106]. Thus, the question arises as to whether airway autoimmune responses promote the TNFSF–TNFRSF axes, leading to induction of ILC2 activity and subsequent promotion of type 2 airway inflammation and corticosteroid resistance. The TL1A–DR3 axis is increased in autoimmune diseases, including RA, IBD, psoriasis, and primary biliary cirrhosis [103,107,108,109]. Serum TL1A levels positively correlate with RA-associated autoantibodies, and TL1A is produced by monocytes stimulated with RA-derived immune complexes in vitro [107]. With respect to asthma, TL1A levels in sputum have been shown to be present in both mild and severe asthmatics, with the highest levels being in prednisone-dependent severe asthmatics with uncontrolled eosinophilia. Interestingly, TL1A levels in sputum from severe eosinophilic asthmatics demonstrated wide heterogeneity where significantly higher levels were found in patients with high sputum autoantibodies compared to those with low autoantibodies [34]. These findings suggest a possible relationship between airway autoimmune responses and the TL1A–DR3 axis in severe asthma. Similar to the RA study showing that monocytes can produce TL1A in response to immune complexes in vitro [107], we have shown that immune complexes derived from sputum of severe asthmatics can induce monocytes to produce TL1A, which is further enhanced in the presence of lipopolysaccharide [34]. Based on these findings, we propose that a possible underlying mechanism of prednisone-dependent severe eosinophilic asthma may be airway autoimmune responses, whereby the presence of airway autoantibodies drives the TL1A–DR3 axis, subsequently activating ILC2 and leading to downstream type 2 inflammation and airway eosinophilia. The proposed relationship between the TL1A–DR3 axis and ILC2 may confer corticosteroid resistance in the prednisone-dependent severe eosinophilic asthmatic population in our study described above. 

It is postulated that the airway cytokine milieu dictates the degree of ILC2 corticosteroid insensitivity, where a combination of alarmin-derived cytokines such as TSLP and TL1A confer corticosteroid insensitivity in ILC2 (Figure 2). Whether other TNFSF–TNFRSF axes are up-regulated in the presence of airway autoantibodies and confer corticosteroid resistance is largely unknown and requires further investigation.

## 6. Conclusions

Substantial evidence has established ILC2 as key regulators of type 2 inflammation in airway disease. Following exposure to environmental stimuli, there is a rapid and sustained accumulation of ILC2 locally within the airways. These resident ILC2 are then influenced by myriad mediators, including alarmins, lipid mediators, and cytokines, resulting in cell activation and production of type 2 cytokines, which promote downstream airway eosinophilia. In particular, ILC2 within the airways appears to contribute to corticosteroid insensitivity in severe prednisone-dependent asthma. This corticosteroid resistance is brought on by the cytokine milieu present within the local environment. Interestingly, a subset of severe prednisone-dependent asthmatics has recently been identified to have airway autoimmune responses, and it has been postulated that this autoimmunity may also contribute to corticosteroid insensitivity. We propose that numerous signaling pathways may be activated in the wake of airway autoimmunity, resulting in downstream corticosteroid resistance of ILC2. However, the exact signaling pathways that are affected by airway autoimmune responses are not well understood. 

What is clear is that numerous TNFSF–TNFRSF axes play a role in promoting ILC2 activity in eosinophilic asthma. These signaling pathways can induce type 2 cytokine production from ILC2 and influence cell proliferation and survival. Animal models demonstrate that knockout of TNFSF–TNFRSF signaling results in attenuation of airway inflammation and AHR, suggesting that these axes have important roles in promoting type 2 inflammation. Interestingly, TNFSF members are associated with several autoimmune diseases, and, recently, we have shown a dichotomy where, in particular, TL1A levels within the airways are increased in severe prednisone-dependent asthmatics with high autoantibody titers. We have further demonstrated that immune complexes isolated from these patients have the capacity to promote TL1A production from monocytes. These findings in combination with the fact that TL1A and TSLP-stimulated ILC2 can confer resistance to corticosteroids identify a possible mechanism as to how airway autoimmunity induces downstream corticosteroid resistance. It is possible that the airway autoimmune responses present in severe asthmatics drive corticosteroid resistance through inducing the production of TL1A or other TNFSF members, which then go on to mediate ILC2 pro-inflammatory activity. We propose that targeting the TL1A–DR3 axis may be a promising therapeutic target for modulating uncontrolled eosinophilia in severe asthmatics with evidence of airway autoimmune responses. Further studies are required to determine what other TNFSF signaling pathways may be influenced by airway autoimmune responses and how these pathways contribute to ILC2 activity and corticosteroid insensitivity in eosinophilic asthma.

## Figures and Tables

**Figure 1 biomolecules-14-00294-f001:**
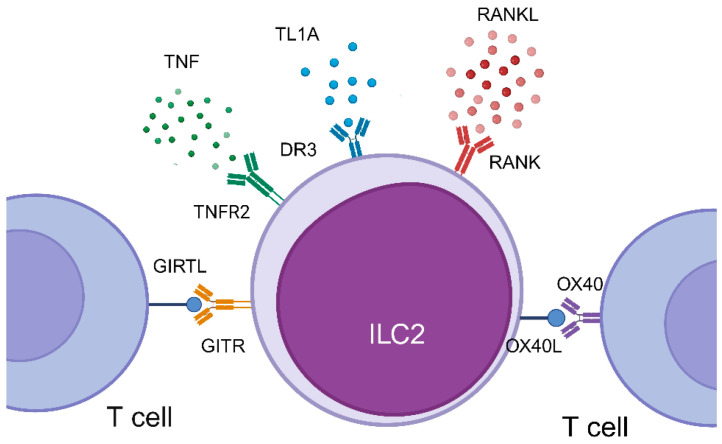
Receptor Expression for TNF Super Family on ILC2. Legend: ILC2 expresses four distinct receptors for TNFSF, including TNFR2, RANK, GITR, and DR3, which bind to their respective ligands/mediators TNF, RANKL, GITRL, and TL1A. Moreover, ILC2 expresses OX40L upon stimulation with IL-33 to activate T cells by binding to OX40 on T cells. DR3 = Death Receptor 3; GITR = Glucocorticoid-induced Tumor Necrosis-Receptor-Related Gene; GITRL = Glucocorticoid-induced Tumor Necrosis-Receptor-Related Gene Ligand; ILC2 = Group 2 Innate Lymphoid Cell; RANK = Receptor Activator of Necrosis Factor Kappa B; RANKL = Receptor Activator of Necrosis Factor Kappa B Ligand; TL1A = Tumor Necrosis Factor-like Ligand 1A; TNF = Tumor Necrosis Factor; TNFR2 = Tumor Necrosis Factor Receptor 2; TNFRSF = Tumor Necrosis Factor Receptor for Super Family; TNFSF = Tumor Necrosis Factor Super Family.

**Figure 2 biomolecules-14-00294-f002:**
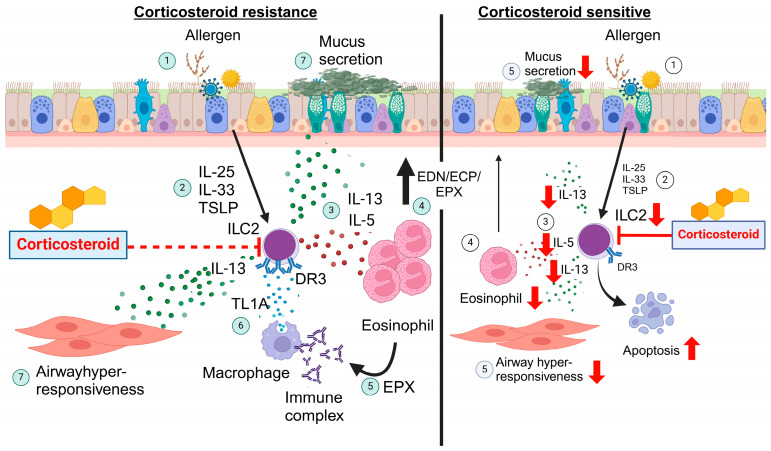
The role of ILC2 in corticosteroid resistant vs. sensitive asthma. Legend: Airway epithelium damage by allergens releases IL-25, IL-33 and TSLP to activate ILC2, inducing airway eosinophilic inflammation. Treatment with corticosteroids has many effects including directly acting on ILC2 to induce apoptosis and suppress airway eosinophilic inflammation (**right**). In contrast, in situations with high airway levels of IL-25, IL-33 and TSLP, autoantibodies to eosinophil-derived granule proteins form immune complexes that stimulate TL1A production by macrophages; TL1A then further promotes type 2 cytokine production from ILC2 via ligation of DR3. Consequently, a potent ILC2-driven airway eosinophilic inflammation, not suppressed by treatment with high-dose corticosteroids, is established. Therefore, in combination with TSLP, TL1A drives ILC2 insensitivity to corticosteroids (**left**). DR3 = Death Receptor 3; ECP = Eosinophil Cationic Protein; EDN = Eosinophil-Derived Neurotoxin; EPX = Eosinophil Peroxidase; TL1A = Tumor Necrosis Factor-like Ligand 1A; TSLP = Thymic Stromal Lymphopoietin.

**Table 1 biomolecules-14-00294-t001:** Summary of evidence regarding relationship between ILC2 and the TNFSF/TNFRSF axes.

Study	TNFSF–TNFRFSF Axis (Model Studied)	Effect of TNFSF–TNFRSF Axis on ILC2
Ogasawara K et al [51].	TNF–TNFR2 (Human)	ILC2 expresses TNFR2 TNF induced further production of IL-5 and IL-13 from human PB ILC2 stimulated with TSLP or IL-33 Comparable TNFR2 expression levels on PB and NP ILC2 from healthy controls and CRS patients
Hurrell, B.P. et al. [55]	TNF–TNFR2 (Human and Mouse)	ILC2 expresses TNFR2 at baseline TNFR2 expression on ILC2 increases 24-72 h post-IL-33 culture TNF induces ILC2 survival and IL-5 and IL-13 production Blockade of TNF/TNFR2 inhibits ILC2-dependent methacholine airway resistance in a mouse model of asthma. TNFR2 signaling induces AHR in Rag2-/- Il2rg-/- mice adoptively transferred human ILC2
Machida K et al. [34]	TL1A–DR3 (Human)	Sputum DR3^+^ ILC2 increases 24 h post-Ag inhalation challenge in subjects with mild asthma Sputum TL1A levels increase 24 h post-Ag IL-2, IL-33 or TSLP stimulation for 24 h up-regulate DR3 expression by enriched PB ILC2, in vitro TL1A induces IL-5 expression by enriched PB ILC2; inhibited by anti-TL1A Together, TL1A + TSLP induce IL-5 expression by ILC2 which was not inhibited by dexamethasone, in vitro
Meylan F et al. [56]	TL1A–DR3 (Mouse)	PB ILC2 constitutively express DR3 at high levels, comparable to CD4^+^ T cells TL1A induces cytokine production by ILC2, including IL-5, IL-6, and IL-13 in vitro Following papain treatment, DR3-KO mice result in significant reduction in lung ILC2 numbers and airway eosinophilia, including peribronchiolar infiltration of eosinophils and goblet cell hyperplasia, compared to wild-type mice
Yu X et al. [50]	TL1A–DR3 (Human and Mouse)	Lung ILC2 numbers increase following intranasal administration of TL1A in DR3-dependent manner in a mouse model of asthma DR3-KO mice have lower ILC2 and type 2 cytokines Human PB ILC2 expresses DR3 Ex-vivo expanded PB ILC2 has up-regulated DR3 expression TL1A stimulates IL-5 and IL-13 production by human ILC2 in vivo
Shafiei-Jahani P et al. [57]	TL1A–DR3 (Human and Mouse)	DR3 is expressed on visceral adipose tissue-derived murine and human PB ILC2 DR3 agonist stimulates naïve murine and human ILC2, inducing increased production of IL-5, IL-13, and GM-CSF in vitro In a murine model of HFD, treatment with DR3 agonist improves glucose tolerance protecting against onset of insulin resistance dependent on ILC2 effector function
Steel H et al. [58]	TL1A–DR3 (Mouse)	Intratracheal administration of TL1A induces a lung muco-secretory signature Mucus production dampens in mice lacking a TL1A–DR3 signal TL1A promotes ILC2-derived IL-13 production for mucus secretion
Momiuchi Y et al. [59]	RANKL–RANK (Mouse)	BM ILC2 highly expresses RANKL, a robust cytokine for osteoclast differentiation and activation Co-culture of BM ILC2 with BMM with IL-7 promotes BMMs differentiation to osteoclasts BM ILC2 stimulated with IL-33 down-regulates RANKL, thereby promoting BMM differentiation to M2 macrophage-like cell through secretion of GM-CSF and IL-13
Ogasawara N et al. [52]	RANKL–RANK (Human)	ILC2 expresses RANK in PB and NP tissue from CRSwNP patients Stimulation with RANK agonist induces type 2 cytokine production from ILC2 in vitro; enhanced with TSLP Co-culture of CD4^+^ T cell and ILC2 from NP results in type 2 cytokine production attenuated by anti-RANKL mAb
Zhang M et al. [53]	GITRL–GITR (Mouse)	ILC2 expresses GITR in lung tissue Stimulation with GITRL results in increased ILC2 in lung in vivo
Galle-Treger L et al. [60]	GITRL–GITR (Human and Mouse)	Human ILC2 expresses GITR Naive and IL-33-stimulated mice express GITR in vivo Human ILC2 stimulated with IL-33 and GITR agonist produces type 2 cytokines in vitro In a murine model of HFD, treatment with GITR agonist improves glucose tolerance and protects against onset of insulin-resistance Following GITR agonist treatment, HFD-fed GITR-KO mice adoptively transferred with ILC2 isolated from wild-type mice exhibits improved insulin resistance compared to those not receiving WT ILC2
Nagashima H et al. [61]	GITRL–GITR (Mouse)	Total ILC2 and activated ILC2 are decreased in lungs of GITR-KO mice compared to wild-type
Drake, LY et al. [16]	OX40L–OX40 (Mouse)	Co-culture of anti-CD3–CD28-stimulates CD4^+^ T cell and ILC2 increases IL-4, IL-5, and IL-13 secretion by CD4^+^ T cell in vitro Treatment with anti-OX40L antibody suppresses production of these cytokines under co-culture of anti-CD3/CD28-stimulates CD4^+^ T cell with ILC2
Halim MA et al. [54]	OX40L–OX40 (Mouse)	Lung ILC2 expresses OX40L following IL-33 stimulation in vivo OX40L on ILC2 attenuates Th2 and Treg expansion and Ag-induced type 2 inflammation
Okuyama, Y et al. [62]	OX40L–OX40 (Mouse)	In an IL-33-induced murine model of melanoma, ILC2 and CD8^+^ T cell infiltrate tumor tissue, inducing an anti-tumor effect, which is dependent on CD8^+^ T cell ILC2 and CD8^+^ T cell in tumors of IL-33-treated mice express OX40L and OX40, respectively Co-culture of OX40+ CD8+ T cells and IL-33-stimulated ILC2 expressing OX40L promotes cell activation and proliferation of CD8^+^ T cell
Wu J et al. [63]	OX40L–OX40 (Mouse)	In a RSV-induced murine model, CD4+ T cell expansion and activation seem to depend upon pulmonary ILC2 Blockade contact between ILC2 and CD4^+^ T cell, significantly diminishes CD4^+^ T cell expansion and cytokine production Co-culture of CD4+ T cells with ILC2 in the presence of anti-OX40L antibody attenuates IFN-γ, IL-2 and IL-13 secretion by CD4^+^ T cell

Abbreviations: Ag = Allergen; BM = Bone Marrow; BMM = BM-derived Monocyte/macrophage Lineage Cell; DR3 = Death Receptor 3; GITR = Glucocorticoid-induced Tumor Necrosis-Receptor-Related Gene; GITRL = Glucocorticoid-induced Tumor Necrosis-Receptor-Related Gene Ligand; GM-CSF = Granulocyte Macrophage Colony-stimulating Factor; HFD = High Fat Diet; ILC2 = Group 2 Innate Lymphoid Cell; KO = Knock Out; mAb = Monoclonal Antibody; PB = Peripheral Blood; RANK = Receptor Activator of Necrosis Factor Kappa B; RANKL = Receptor Activator of Necrosis Factor Kappa B Ligand; RSV = Respiratory Syncytial Virus; Treg = Regulatory T cells; TL1A = Tumor Necrosis Factor-like Ligand 1A; Th2 = Type 2 Helper T cells; TNF = Tumor Necrosis Factor; TNFSF = Tumor Necrosis Factor Superfamily; TNFRSF = Tumor Necrosis Factor Receptor Super Family; TNFR2 = Tumor Necrosis Factor Receptor 2.

## Data Availability

Not applicable.

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
