# Peer review of "TNF Superfamily and ILC2 Activation in Asthma"

_biomolecules, 2024, doi:10.3390/biom14030294_

Round 1

Reviewer 1 Report

Comments and Suggestions for Authors

General comments

This review is an extensive review of publications related to innate lymphoid cells (type 2). The authors should consider removing text that relates to pathologies that are not linked to the respiratory system such as diabetes mellitus and bone metabolism. The author frequently draws conclusions that are too strong. The role of ILCs in human asthma remains speculative and based on correlative evidence. Causality has not been established. TNF has not been established through clinical trials of biologics. Early studies failed to implicate it in asthma. Thus, a more conservative approach should be adopted. The text in places is quite dense and would benefit from more interpretation. Many sentences have three or more ideas within them, making reading difficult. There is much focus on immunology but not so much on airway function. If there is a lack of data then it would be useful to acknowledge. Is there a role for ILC subsets in non-eosinophilic phenotypes of asthma which represent almost 50% of asthmatics.

Specific comments

Line 49. The title suggests that asthma is being addressed but there is a lot of text on nasal polyps and CRS. Perhaps the title of this subsection should be revised.

Line 68. In vitro should not be hyphenated.

Line 90. Please specify again the two conditions you are referring to.

Line 101. Should read “comparable to or lower than..”.

Line 110. How was IL-33 measured? In sputum, BAL fluid?

Line 118. What is the direction of the correlation? Direct or inverse?

Line 122. It would be useful to define severe asthma. Is it based on steroid resistance?

Line 132. “are detected in …”.

Line 170. Please interpret the significance of CD45RO and RA on ILCs.

Line 218. The section should make clear what is actually known and what is speculation.

Lines 229 to 239. There is no reference provided for the data discussed.

Line 285. What is specifically meant by homeostasis here?

Line 288. “the regulatory T cell…”.

Line 321. It is not clear what has been observed. It would better to be explicit.

Lines 342 to 346. This text seems like a digression from the principal focus of the paper.

Line 346. “a novel therapeutic option…”.

Line 358. There is no need to abbreviate antigen/allergen in this way.

Line 387. Delete malignancy. Bone metastases define the prowess as malignant.

Line 396. “plays a role”.

Line 446. Is it suggested here that eosinophilia is protective against diabetes?

Line 466. Do you mean OX40/OX40L interaction?

Line 490 – How often do severe asthmatics have autoimmune signatures? Are they all associated with eosinophilia? Some clarification would be helpful.

Line 494. Should read “severe asthmatics with autoimmunity …”.

Line 508. Avoid abbreviations like aAb. This is a non-standard abbreviation.

Line 517. Delete “infective”.

Line 525. Is there evidence of expanded ILC2 cells in the severe autoimmune asthmatic phenotype?

Line 536. Please explain what is meant by dichotomy here.

Figure legends merit a more detailed description.

Comments on the Quality of English Language

The language is satisfactory but the style of writing makes reading difficult. There needs to be more interpretation and avoidance of sentences with too many concurrent ideas/facts.  

Author Response

Dear Reviewer 1 thank you for your very detailed review of our manuscript. Please find attached the reviewer comments to your questions and suggestions.

Reviewer 2 Report

Comments and Suggestions for Authors

This is a very well written review article Dr Matsuyama et al discussing evidence that support a relationship between ILC2 and the tumor necrosis factor superfamily (TNFSF) and its receptors (TNFRSF) axis in eosinophilic asthma. 

They also discuss a potential role for the this axis in severe asthma with the precense of airway autoimmune responses.

Minor comments:

1. Figure 1 line 263 seem like the legend text is cut.

2. Table 1 should be organized horizontal instead of vertical. The dots before the statments (effect of TNFSF/TNFRSF Axis on ILC2 should be lined up.

3. Line 512 autoantibodies should be abbreviated (aAbs)

Author Response

This is a very well written review article Dr Matsuyama et al discussing evidence that support a relationship between ILC2 and the tumor necrosis factor superfamily (TNFSF) and its receptors (TNFRSF) axis in eosinophilic asthma. 

They also discuss a potential role for the this axis in severe asthma with the precense of airway autoimmune responses.

Minor comments:

  1. Figure 1 line 263 seem like the legend text is cut.

Response: As you suggested, we deleted “the TNF super family” in the legend in line 247, because it is listed in the abbreviations in line 257.

  1. Table 1 should be organized horizontal instead of vertical. The dots before the statements (effect of TNFSF/TNFRSF Axis on ILC2 should be lined up.

Response: As you pointed out, we lined up the dots before the statements. On the other hand, it is difficult to compose the table horizontally because it would not fit in the space available in the manuscript. Therefore, we left the table.

  1. Line 512 autoantibodies should be abbreviated (aAbs)

Response: In the major review, one of the reviewers indicated that aAbs was a non-standard abbreviation. Because we agreed with this opinion, we retained this expression.

Reviewer 3 Report

Comments and Suggestions for Authors

Takahiro Matsuyama et al. present here in this review on eosinophilic asthma evidence that support a relationship between cellular actor ILC2 and members of the TNF super family.

After presenting ILC2 and their known roles in eosinophilic asthma, authors review further ILC2 impact on corticosteroid resistance, before reviewing roles of TNFRSF on ILC2 biology. Lastly, they comment on airway autoimmune responses. The table and 2 figures illustrate well the content presented.

Major comments:

In the introduction, please define briefly the term alarmin.

line56, add some examples of alarmin cytokines as examples were given for the other mediators.

line342-346, as line 440-442 and 448: references to another pathology, diabetes, insulin response: it is a review on eosinophilic asthma and it is out of topic, I recommend to delete these parts.

On other places throughout the manuscript, the authors already explain a lot on biology not related at all with asthma or even Th2 responses, it is done to present what is known on the specific receptor, so it is supporting the topic.

But I don’t see why the authors talked about diabetes.

Figure 2 needs a little bit more clarity on the corticoid effect or the lack thereof.

steps are numbered, but drug effects could be presented better.

Minor comments:

line18: complete the sentence, are the authors meaning: promote ILC2 ‘activity’.

Figure 1 legend : the beginning is cut out, delete end of line 270.

TNFSF, TNFRSF are not in the figure, please discard these names from the legend.

References format needs revision to fit the journal instructions.

Author Response

Dear Reviewer 2 thank you for your very detailed review of our manuscript. Please find attached the reviewer comments to your questions and suggestions.
